# Methodology to obtain highly resolved SO₂ vertical profiles for representation of volcanic emissions in climate models

Oscar S. Sandvik[1], Johan Friberg*[1], Moa K. Sporre[1], Bengt G. Martinsson[1]

[1]Department of Physics, Lund University, Lund, 22100, Sweden

*Correspondence to*: Johan Friberg (johan.friberg@nuclear.lu.se)

**Abstract.** In this study we describe a methodology to create high vertical resolution $SO_2$ profiles from volcanic emissions. We demonstrate the method's performance for the volcanic clouds following the eruption of Sarychev in June 2009. The resulting profiles are based on a combination of satellite $SO_2$ and aerosol retrievals together with trajectory modelling. We use satellite-based measurements, namely lidar back-scattering profiles from the Cloud-Aerosol Lidar with Orthogonal Polarization

(CALIOP) satellite instrument to create vertical profiles for $SO_2$ swaths from the Atmospheric Infrared Sounder (AIRS) aboard the Aqua satellite. Vertical profiles are created by transporting the air containing volcanic aerosol seen in CALIOP observations using the dispersion model FLEXPART, while preserving the high vertical resolution by using the potential temperatures from the MERRA-2 meteorological data for the original CALIOP swaths. For the Sarychev eruption, air tracers from 75 CALIOP swaths within 9 days after the eruption are transported forwards and backwards, and then combined at a point in time when

AIRS swaths cover the complete volcanic $SO_2$ cloud. Our method creates vertical distributions for column density observations of $SO_2$ for individual AIRS swaths, using height information from multiple CALIOP swaths. The resulting dataset gives insight to the height distribution in the different sub-clouds of $SO_2$ within the stratosphere. We have compiled a gridded high vertical resolution $SO_2$ inventory that can be used in Earth system models, with vertical resolution of 1 K in potential temperature, $61\pm56$ m, or $1.8\pm2.9$ mbar.

## 1 Introduction

Volcanism can affect the climate by increasing aerosol levels in the stratosphere (Robock, 2000). The strongest effect is seen from the eruptions that emitted the largest amounts of $SO_2$ to high altitudes, such as Mt. Pinatubo in 1991 which is estimated to have decreased global average surface temperatures by several tenths of a degree Celsius (Kremser et al., 2016; Canty et al., 2013). A significant effect from moderate sized eruptions that reach into the stratosphere has also been reported (Vernier et

et al., 2011a;Andersson et al., 2015;Ge et al., 2016). The stratospheric aerosol from these eruptions can have a climate effect if enough $SO_2$ is released into the stratosphere. This climate effect was underestimated in the CMIP5 simulations since the simulations did not take into account the increased stratospheric volcanic aerosol loadings from moderate sized eruptions (Solomon et al., 2011;Santer et al., 2014).

The duration of the effect of volcanism on climate is highly dependent on the altitude where the $SO_2$ is released (SPARC, 2006). Aerosol in the stratosphere can remain there for several years depending on the injection height and through which branch of the Brewer-Dobson circulation it is transported (Butchart, 2014;Friberg et al., 2018). Since transport mainly occurs on surfaces of constant potential temperature, the stratosphere is usually layered into intervals of potential temperature with the lowermost stratosphere (LMS) being bounded by the tropopause and the 380 K potential temperature surface. And while aerosol can reside in the stratosphere for a long time, once the aerosol particles descend below the tropopause they will rapidly be removed from the atmosphere by wet deposition. Global climate models use $SO_2$ emission observations from satellite based instruments to estimate the climate impact of volcanic eruptions. One conundrum is to provide $SO_2$ concentrations for climate modellers at accurate heights (Timmreck et al., 2018).

Volcanic eruptions intermittently add aerosol and aerosol precursor gases to the stratospheric background. This stratospheric background aerosol mainly consists of water-soluble sulphate, organics, black carbon, and extra-terrestrial material (Murphy et al., 1998;Martinsson et al., 2009;Friberg et al., 2014;Sandvik et al., 2019). The gas precursors for the sulphate in the stratosphere are mostly $SO_2$ and carbonyl sulphide (OCS), with OCS being released from oceans and anthropogenic sources (Kremser et al., 2016). The Brewer-Dobson circulation seasonally transports aerosol from the overlying layers down to the LMS (Martinsson et al., 2019). Wildfires also contribute with aerosol particles to the stratosphere (Fromm et al., 2000;Khaykin et al., 2018;Peterson et al., 2018;Kablick et al., 2020). There is a seasonal aerosol layer called the Asian Tropopause Aerosol Layer (ATAL) in the region 5-105°E and 15-45°N that also contribute to the background stratospheric aerosol (Vernier et al., 2015).

Satellite instruments can be used to measure the stratospheric aerosol and trace gases from volcanism (Kremser et al., 2016). The satellites in the A-train satellite constellation pass over the same locations near simultaneously (Stephens et al., 2002). Two of these satellites, Cloud-Aerosol Lidar and Infrared Pathfinder Satellite Observation (CALIPSO) and Aqua, are used in this study. CALIPSO has the CALIOP lidar instrument on-board which provides backscattering height profiles with a vertical resolution of 60 m in the lower stratosphere (Winker et al., 2010). Aqua carries the Atmospheric Infrared Sounder (AIRS) instrument. It has been used to measure vertical column densities of $SO_2$ over wide areas with high spatial resolution but with limited vertical resolution (Prata and Bernardo, 2007).

Since most $SO_2$-sensors are passive satellite instruments, the height information needs to be estimated indirectly. Due to interference from water vapour at lower altitudes, the vertical column densities of $SO_2$ from AIRS is representative for the upper troposphere and the lower stratosphere. The focus of this study is aerosol layers found above the tropopause in the CALIOP datasets, which is where climate-impacting volcanic aerosol is situated. CALIOP and AIRS are part of the large family of satellite instruments measuring aerosol and $SO_2$ (Thies and Bendix, 2011). Another instrument is the Infrared Atmospheric Sounding Interferometer (IASI), aboard the METOP satellite, which has been used to infer a plume altitude and

SO$_2$ levels simultaneously from high-spectral resolution measurements (Carboni et al., 2016). However, volcanic aerosol can be injected into several altitude layers over the same location and CALIOP can readily detect these distributions whereas IASI retrieves only a single altitude per pixel. Wu et al. (2017) used the AIRS SO$_2$ sensor together with trajectory computations, validating the obtained profile by aerosol measurements from the satellite-based Michelson Interferometer for Passive Atmospheric Sounding (MIPAS) with the vertical resolution 3 – 4 km (Günther et al., 2018).

This work proposes a new method for retrieving height profiles of the SO$_2$ observed by passive instruments. The method uses AIRS for vertical column densities of SO$_2$ and then uses vertical profiles from CALIOP swaths with a vertical resolution of 0.06 km to create vertical profiles of SO$_2$ concentration for the emissions of a volcanic eruption. The FLEXible PARTicle dispersion model (FLEXPART) (Pisso et al., 2019) is used to transport the horizontally thin CALIOP observations to the time and location of the SO$_2$ swaths. This approach enables us to use height information from multiple CALIOP swaths for each AIRS swath giving a more complete view of the SO$_2$ clouds vertical profiles, see Fig. 1. In order to do this, we have made the assumption that SO$_2$ and aerosol particles are co-located and have the same height profile.

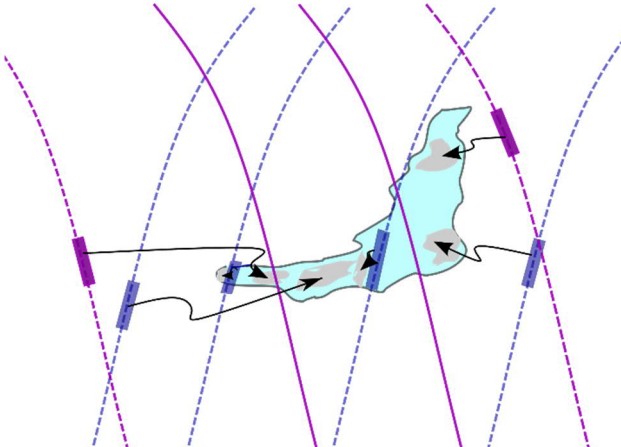

Figure 1. Concept sketch of the use of many CALIOP swaths to determine the height of an SO$_2$ cloud (shown in light blue). Vertical profiles from the daytime CALIOP orbital tracks retrieved simultaneously as the AIRS swaths (purple) is used together with data from CALIOP night-time orbital tracks (blue dashed lines) and daytime CALIOP orbital tracks from other instances (purple dashed lines). The thick blocks segments on the orbital tracks are the segments where volcanic aerosol is detected. Since the volcanic SO$_2$ cloud is subject to transport, the height information from the non-concurrent swaths is transported using FLEXPART (black arrows) to give a more complete picture of the altitude of the SO$_2$ cloud. The grey areas are where FLEXPART trajectories contribute with SO$_2$ cloud altitude information.

## 2 Eruption, instruments and models

This section will provide background information on the Sarychev 2009 eruption, the AIRS and CALIOP instruments, FLEXPART and supporting datasets.

5    The altitudes of the Sarychev 2009 eruptions are difficult to determine and the complete set of eruptions is therefore a good candidate to develop and demonstrate our method on. The Sarychev volcano erupted several times over the course of five days and injected $SO_2$ at various altitudes, creating a complex $SO_2$ vertical profile. The 2009 Sarychev eruption started on the 11[th] of June by mainly emitting ash, and on the 14[th] of June there was an isolated eruption that reached an estimated altitude of around 21 km (Levin et al., 2010). The 15[th] of June had the highest number of eruptions reaching above 6 km (Levin et al., 2010) and most of the emitted $SO_2$ was released on this day according to Rybin et al. (2011). A second large plume was observed on June 16 using IASI (Haywood et al., 2010). From the 17[th], no stratospheric $SO_2$ injections was found. The total emitted $SO_2$ mass of the eruption has been reported to be $1.2\pm0.1$ Tg by (Haywood et al., 2010), 0.6 Tg by Carboni et al. (2016) and 0.9 Tg by Berthet et al. (2017). A total $SO_2$ mass of 0.9 Tg is also reported in Fig. 8 in Carn et al. (2016).

### 2.1 AIRS

15    The Atmospheric Infrared Sounder (AIRS) was launched on-board the Aqua satellite in 2002 with the purpose to improve weather predictions and provide measurements of gases important for our understanding of the climate (Chahine et al., 2006). AIRS measures infrared light from the atmosphere in 2378 channels between 650 and 2665 $cm^{-1}$ with a high spectral resolution (Chahine et al., 2006). The horizontal resolution of AIRS pixels in a product is $15 \times 15$ $km^2$ at nadir and $18 \times 40$ $km^2$ at the edges of a swath (Prata and Bernardo, 2007).

By using the channels sensitive to $SO_2$ in the spectrum covered by AIRS, $SO_2$ column densities are retrieved (Carn, 2005;Prata and Bernardo, 2007). In this study we used $SO_2$ data provided by Dr. Fred Prata, with methods described in Prata and Bernardo (2007). Their dataset is based on a least squares fit between results from radiative transfer simulations and the observed spectrum from AIRS. They reported that the accuracy of the retrieval scheme was $\pm6$ D.U. (1 D.U. = $2.9\cdot10^{-5}$ kg $SO_2$ $m^{-2}$), but

25    in a case with little water vapour interference and good background conditions the accuracy has been estimated to be as good as $\pm3$ D.U. (Eckhardt et al., 2008). A collection of AIRS swaths covering the Sarychev $SO_2$ cloud is shown in Fig. 2. By using this collection of AIRS swaths, the complete $SO_2$ emissions are studied in this paper.

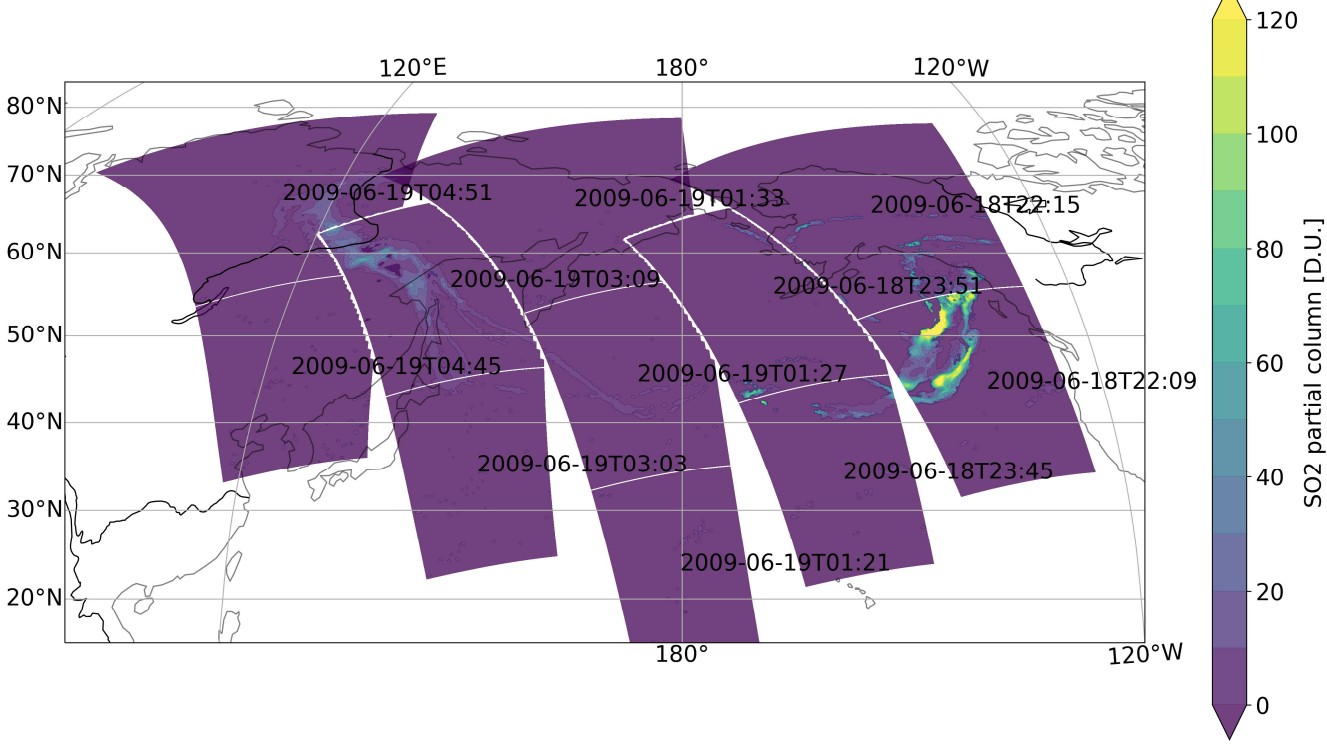

**Figure 2: SO₂ partial column densities from AIRS swaths on the 18th and 19th of June. The total SO₂ mass is 1.09 Tg with no location counted more than once.**

## 2.2 CALIOP

5   CALIOP is a polarization sensitive lidar instrument, and was launched on-board the CALIPSO satellite in 2006 (Winker et al., 2007) . CALIOP has been used extensively to track the height of volcanic aerosols (Kristiansen et al., 2010;Vernier et al., 2011b;Andersson et al., 2015;Friberg et al., 2018;Sandvik et al., 2019). In Fig. 3, we show a CALIOP swath containing volcanic aerosol from the Sarychev 2009 eruption. In this study we have used the level 1b product from version 4-10 (Kar et al., 2018;Getzewich et al., 2018), which has a horizontal resolution of 300 m and is the raw product.

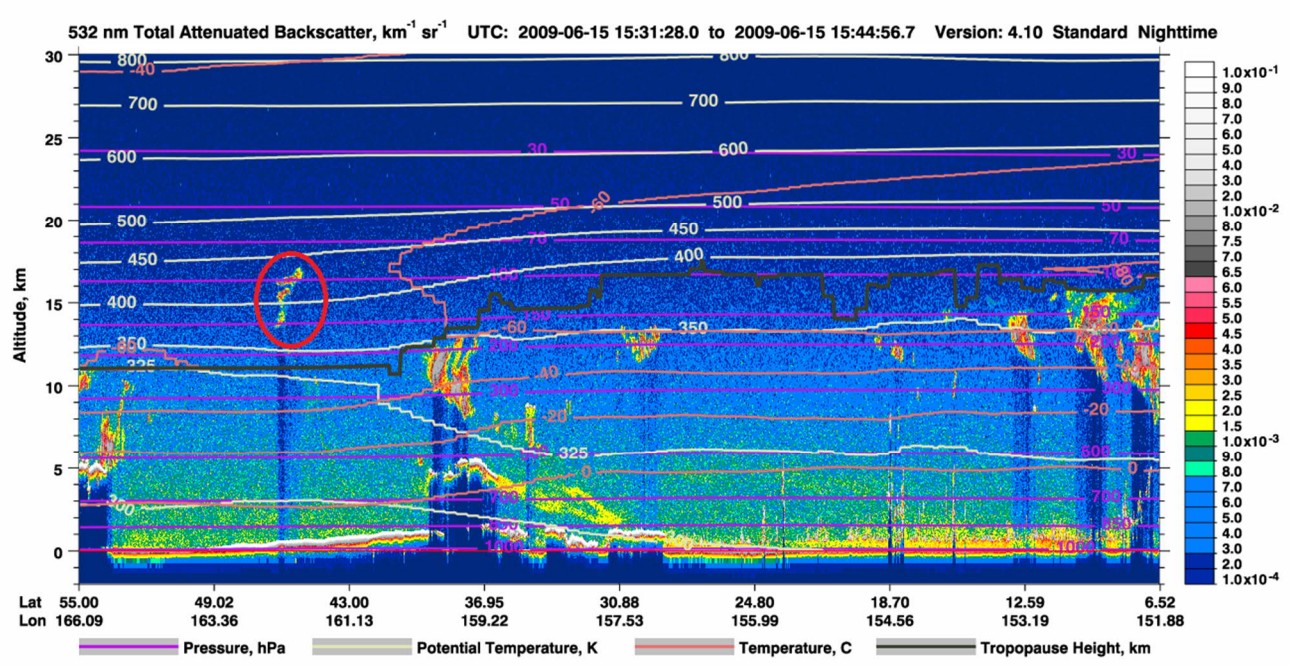

**Figure 3: A CALIOP swath containing volcanic aerosol at latitude 46°N and altitudes between 13 and 17 km (red circle). This CALIOP swath is also used in Fig. 5. This figure is part of the CALIOP browse images from NASA.**

While SO$_2$ data from passive satellite sensors can have an estimated altitude resolution of 1-2 km (Carboni et al., 2012), CALIOP has higher vertical resolution. The vertical resolution is highest at lower altitudes: 30 m between the surface and 8.2 km altitude, 60 m between 8.2 and 20.2 km altitude, 180 m between 20.2 and 30.1 km altitude, and 300 m above 30.1 km (Winker et al., 2010), i.e. more than an order of magnitude better than satellite borne SO$_2$ sensors in the lower stratosphere.

Since CALIPSO and AIRS at the time of the 2009 Sarychev eruption were in the A-train satellite constellation, CALIOP passes over the same air masses as AIRS. However, due to the narrow footprint of the CALIOP laser beam, it measures a thin slice of the AIRS footprint.

In this study we have used the attenuated backscattering from CALIOP to calculate the scattering from volcanic aerosol

particles. From the total attenuated backscattering measured by CALIOP, $\beta_{tot}$, we define aerosol backscattering as:

$$\beta_{aer} = \beta_{tot} - \beta_{mol}$$

Where we calculated the molecular backscattering, $\beta_{mol}$, from the meteorological parameters from the MERRA-2 model provided with the 4-10 version of the CALIOP data (Kar et al., 2018). In order to have a more height invariant variable for aerosol scattering, we also define scattering ratio as:

$$SR = \frac{\beta_{tot}}{\beta_{mol}}$$

In this study, we use individual CALIOP swaths to observe the early transport of volcanic aerosol layers that clearly contrast the background aerosol. Since CALIOP did not produce any data on the 13[th] of June, our selection of swaths starts on the 14[th] of June when the first aerosol layers are observed. The aerosol layers become less dense in individual CALIOP swaths as time progresses and the last day for our selection of swaths is the 22[nd] of June. Further into our method, this time interval has the added benefit of avoiding long transport times with the FLEXPART model. Due to the rapid removal of aerosols in the troposphere, we have solely focused on stratospheric aerosol layers in this study.

To make sure that not a large portion of the $SO_2$ mass in the AIRS swath collection is located in the upper troposphere, we performed manual inspection of co-located CALIOP swaths to determine the height of the volcanic clouds. In half of the AIRS swaths in our time period, the $SO_2$ was solely located in the stratosphere. Within the other half of the AIRS swaths, the volcanic aerosol layers were mostly located in the stratosphere with minor aerosol layers in the underlying troposphere. Thus, the AIRS swaths shown in Fig. 2 contain volcanic aerosol in the stratosphere and very little in the troposphere. In the Swaths passing over the region close to the volcano there is also volcanic aerosol further down in the troposphere, close to the ground. However, the AIRS $SO_2$ measurements are less sensitive at these altitudes due to water vapour interference.

In the chosen time interval, we found 75 CALIOP swaths (see Table S1 for full list). These swaths contained stratospheric aerosol layers that were clearly separated from the background aerosol and not in contact with ice clouds. The CALIOP swaths identified in this way contain most of the stratospheric aerosol layers from the Sarychev eruption in this interval.

To check if there were ash or ice altering the measured height profiles within these stratospheric aerosol layers, we used the ratio between perpendicularly polarized and total backscattering, called volume depolarization ratio, and the ratio between the total backscattering of the 1064 nm and 532 nm channels, called colour ratio. Previous CALIOP classification algorithms in version 3 would misclassify fresh stratospheric volcanic aerosol as ice clouds when the stratospheric aerosol had depolarization ratios over 0.03 (Liu et al., 2019). For ice containing pixels, the depolarization ratios would mostly be above 0.3. This threshold was used by Khaykin et al. (2018). The threshold is corroborated by Fig. 4 in Liu et al. (2019), where aerosol layers become increasingly scarce when the depolarization ratio exceeds 0.3. These pixels were instead classified as clouds in their study. In the aerosol layers of the 75 CALIOP swaths, the particle depolarization ratios were 0.1-0.3. The depolarization ratios are lower in the later swaths, indicating formation and condensation of sulphate. Sedimentation of ash particles could alter the vertical profile of the aerosol, whereas the submicron sulphate particles have negligible sedimentation rate during the short time-period in our study (Martinsson et al. 2005). This may not be the case for more ash-rich eruptions. In such a case, the ash would be unevenly distributed vertically inside the layers since the large particles would settle more quickly than smaller ones, inducing

a vertical gradient in the depolarization ratio (Vernier et al., 2016). We investigated the vertical distribution of the depolarization ratio and colour ratio in relation to the scattering ratio for the 75 CALIOP swaths. We found no evidence for vertical inhomogeneity in this study of the Sarychev 2009 eruption. This indicates that $SO_2$ and aerosol particles have the same vertical distribution. This co-location means that we have detailed height profiles of how the $SO_2$ is distributed, albeit in thin
slices in each CALIOP-swath. These thin slices of height profiles from all 75 CALIOP swaths were transported using the dispersion model FLEXPART (see next section) to the times of individual AIRS swaths so that the AIRS swaths get matching height profiles.

## 2.3 FLEXPART

FLEXPART, full name FLEXible PARTicle dispersion model, is a trajectory and dispersion model with Lagrangian dynamics
(Pisso et al., 2019). By being Lagrangian, FLEXPART tracks each individual air tracer particle's position instead of calculating box quantities. This study uses FLEXPART Version 10.4. The model was originally developed to track radioactive particles (Pisso et al., 2019) but has since been used for many other types of studies, e.g. to track volcanic clouds (Eckhardt et al., 2008;Kristiansen et al., 2010). Transport with FLEXPART produces results in good agreement with both transport simulated by the Norwegian Earth System Model (Cassiani et al., 2016), and to observational data (Groot Zwaaftink et al., 2018;Langford
et al., 2018).

In this study we released air tracer particles in FLEXPART from each aerosol layer observed by CALIOP, with each FLEXPART release corresponding to a single pixel in the CALIOP data's aerosol layers. In order to manage this and to simulate FLEXPART both forwards and backwards starting from each CALIOP swath we created one "RELEASES"-file,
specifying where and how many particles should be released into the model, and two "COMMAND"-files, specifying how the model is run and for how long, for each CALIOP swath. Each RELEASES-file contained around 95 000 air tracer particles.

## 2.4 Meteorological data

We used ERA5 meteorological data in the FLEXPART simulations. ERA5 is the latest reanalysis product from ECMWF, replacing ERA-interim (Hersbach et al., 2020). The preparation of ERA5 data for input into FLEXPART was done with
flex_extract_v7.1 (Philipp et al., 2020) with a hourly $1°\times1°$ resolution on 137 vertical levels. In the CALIOP data, the tropopause and potential temperature levels come from the MERRA-2 model (Kar et al., 2018). Using the supporting meteorological dataset for the CALIOP swaths, our final vertical profiles have a resolution of 1 K in potential temperature. Converted to geometric altitude and pressure this corresponds to $61\pm56$ m and $1.8\pm2.9$ mbar (mean $\pm$ 1 standard deviation) for the altitude range of the volcanic sub-clouds in the AIRS swaths in Fig.1.

## 3 Retrieval of vertical distributions

In this section, we will describe and show results from each step in this method to obtain vertical profiles for $SO_2$ emissions. We use the Sarychev eruption in June 2009 to develop our method and the $SO_2$ observations shown in Fig. 2 since this is AIRS's most complete coverage of the volcanic $SO_2$ clouds. We use the satellite-based lidar CALIOP for the vertical profiles of particles co-located with $SO_2$. CALIOP observations only cover a small portion of the $SO_2$ emissions at various times. Therefore, CALIOP observations inside a four-day span before and after the observations in Fig. 2 were transported to the times of the swaths in that figure using FLEXPART simulations, see Fig. 1. An overview of our method and how it is presented in the different sections is shown in Fig. 4.

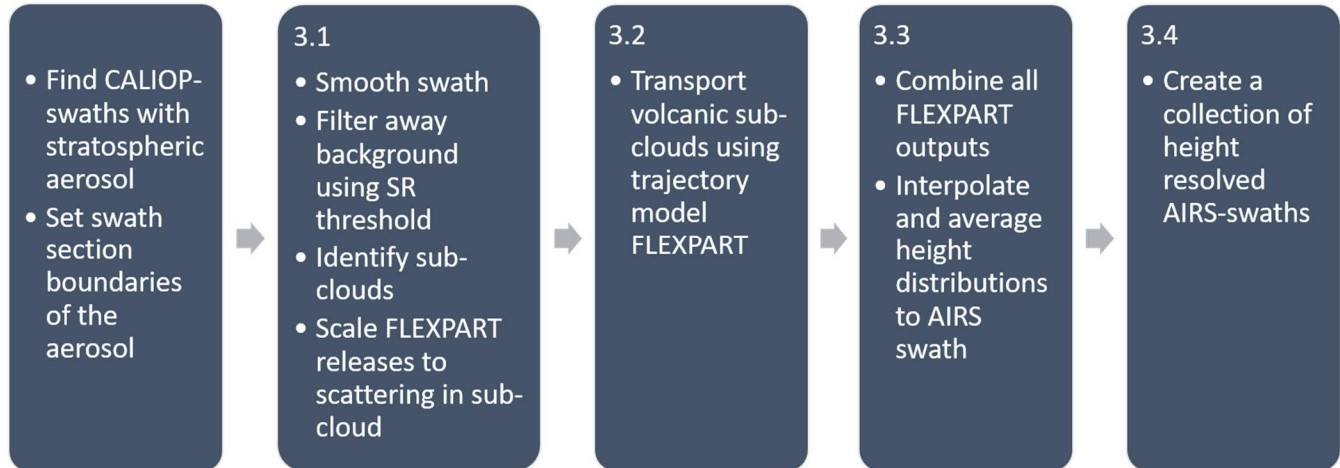

**Figure 4: Flowchart of our method.**

### 3.1 Preparation of CALIOP data for FLEXPART release

Starting from a section of a CALIOP swath that contains volcanic aerosol layers, see Fig. 5a and the solid blocks in Fig 1, information that can be turned into a FLEXPART release is extracted and prepared. Smoothing is applied to the CALIOP data to reduce noise. This is done by applying a moving mean on the data, see Fig. 5b. A scattering ratio threshold (10 for night swaths and 15 for day swaths due to more noise in day data) is used to separate the volcanic cloud from the background. With the background filtered away all that remains in the CALIOP data is separated volcanic sub-clouds, see Fig. 5c. In order to keep track of these sub-clouds throughout our analysis, we grouped the grid-cells belonging to each sub-cloud and put id-labels on them, see Fig. 5d. These id-labels were used in the later analysis to keep track of which FLEXPART particles come from which sub-cloud. During manual inspection of the selected sub-clouds, sub-clouds with a high likelihood of containing ice and ash were removed. These ice and ash sub-clouds contained only 2.1 % of the total light scattering observed by CALIOP from the clouds classified as volcanic. Hence, their removal did not affect our final results.

With the sub-clouds now clearly identified, we want to convert the backscattering into FLEXPART air tracer particles. This is done by swath-wise scaling the number of released FLEXPART air tracer particles by the aerosol backscattering ($\beta_{aer}$) in a pixel, making $\beta_{aer}$ the relative strength of a FLEXPART release. Thus, FLEXPART air tracer particles are released for each pixel in a sub-cloud in the CALIOP data, and at the time, latitude, longitude, and geometric altitude of the pixel. Around 95

000 air tracer particles were released for each FLEXPART simulation. i.e. for each CALIOP swath (except for the two swaths that were divided into two separate simulations).

For the released FLEXPART air tracer particles in each CALIOP pixel, the potential temperature is stored. Potential temperature is a robust height coordinate in the stratosphere since the air transport normally follows isentropes, which are

10 usually not aligned with geometric altitudes. Therefore, the potential temperature is used later as the vertical coordinate for making combined height profiles out of the FLEXPART simulations that cover the locations and times of the AIRS observations. Dense super-micron aerosol layers such as those from Mt Pinatubo 1991 can be warmed through absorption of radiation causing cross-isentropic transport, so-called self-lofting. It is unknown to what extent self-lofting is important for moderate sized eruptions but a study of Nabro (Fairlie et al., 2013), another moderate sized eruption, found the effect to be

minor (max 0.3 K/day). Self-lofting is not captured by our method, but its impact can be assumed negligible for Sarychev given the low rates in Fairlie et al. (2013) and the short time-span in our study.

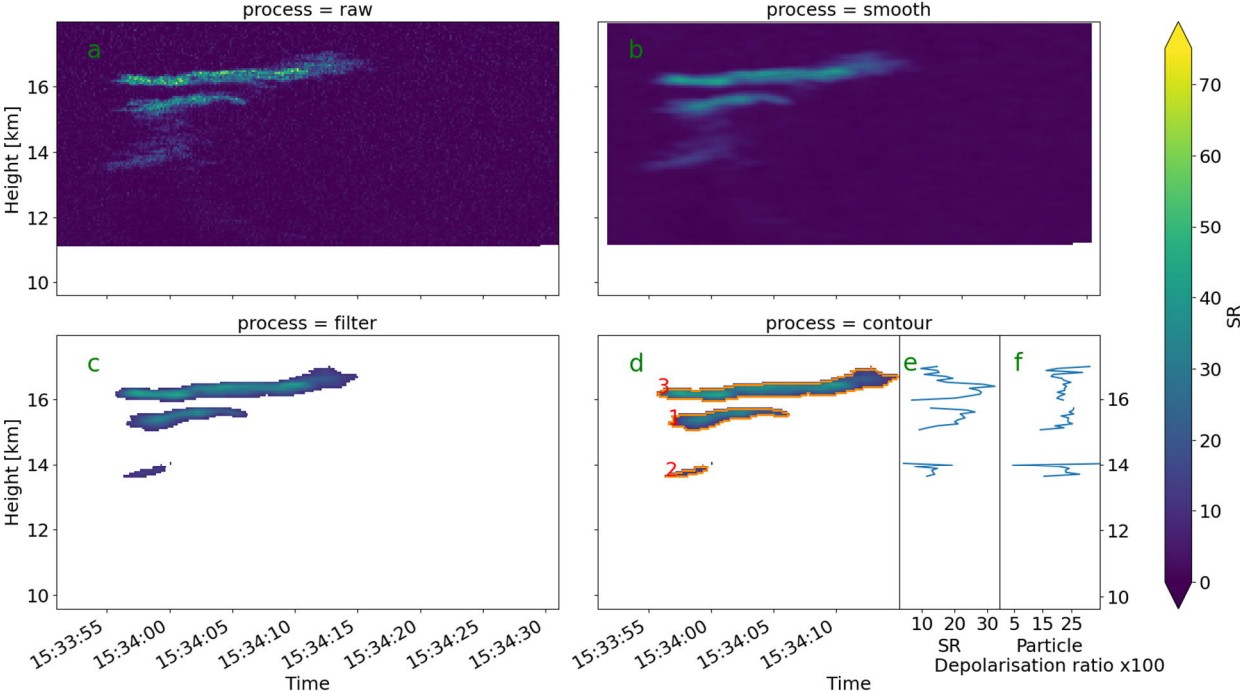

**Figure 5: Example of how a volcanic cloud is located in a CALIOP swath and how it is prepped for RELEASES, a) the calculated scattering ratio from a level 1 CALIOP swath with pixels below the tropopause filtered away (white), b) the scattering ratio after**

20 **smoothing has been applied, c) the smoothed scattering ratio after the scattering ratio threshold has been applied, d) sub-clouds**

identified by computer vision. The assigned id-numbers are used to track the individual sub-clouds when FLEXPART simulates them simultaneously. Additionally, in e) and f), the vertical profiles of SR and depolarization ratios of the average backscattering in the sub-clouds are shown. The steep shifts near the edges are due to few pixels with weak signal. Note that the profile in f) is otherwise fairly homogeneous. The swath is CAL_LID_L1-Standard-V4-10.2009-06-15T15-31-31ZN (the same as in Fig. 3).

## 3.2 Air tracer transport using FLEXPART

The released particles from CALIOP were transported with FLEXPART forwards or backwards in time until the time of the AIRS swaths in Fig. 2 (see the black arrows in Fig 1). Hence, we have collected and transported all the vertical information from the 75 CALIOP swaths to the times of each AIRS swath. An advantage of making one FLEXPART simulation for each CALIOP swath is that it increases the possible number of air tracer particles from every CALIOP swath. One swath contained aerosol layers at two locations separated by a large distance and could not be enclosed by boundaries without also including a large chunk of noise. The aerosol at the two locations were therefore prepared and run separately. Our approach enabled us to do all FLEXPART simulations on an ordinary personal computer. The alternative would have been to make a single FLEXPART simulation containing all CALIOP swaths, where program restrictions would limit the number of air tracer particles per CALIOP swath.

From a technical point of view, the procedures in the paragraph above were implemented by creating two FLEXPART COMMAND-files for each RELEASES-file: One COMMAND-file tells FLEXPART to simulate in the forward time direction and one in the backwards direction (See supplementary material for an example COMMAND-file). For forward (backward) simulations, FLEXPART starts at the closest hour before (after) the first (last) release in the corresponding RELEASES file and stops after the last (before the first) AIRS swath in Fig. 2. All in all, 152 FLEXPART simulations were made for a total of 836 simulated days.

While FLEXPART's internal calculations have no gridded resolution, the produced output data were placed on a grid. In our study, we chose $0.5°E \times 0.5°N$ resolution for the northern hemisphere. The southern hemisphere was ignored since the cloud from Sarychev stayed far from the equator in the first weeks after the eruption. The high horizontal resolution of the FLEXPART output was used because both CALIOP and AIRS have high horizontal resolutions. The chosen FLEXPART output time intervals were set to 30 minutes and are therefore within 15 min before or after an AIRS swath, i.e. there is only minor possible misalignment in time between the satellite and model data.

We used FLEXPART to transport the data in the CALIOP swaths horizontally through time. Since CALIOP already has the highly resolved height information, we rely on that observational data for vertical coordinates while FLEXPART produces the horizontal transport We therefore set only one single vertical coordinate for the FLEXPART output. This approach has the advantages that our method becomes independent of possible errors in vertical transport in FLEXPART and allow us to increase the resolution of the other output coordinates.

### 3.3 Height profile of an AIRS swath

After FLEXPART has simulated every CALIOP swath, the numerous output files were combined. Vertical profiles were created for each AIRS pixel by spreading the SO2 mass in each AIRS pixel over a range of potential temperature bins. The amount of SO2 in each bin was computed by weighting the FLEXPART data with the aerosol backscattering data from 5 CALIOP.

No chemistry is used during the transport or for the preparation of data before FLEXPART. Therefore, one unit of $\beta_{aer}$ represents the same amount of $SO_2$ regardless of CALIOP swath time. The time it takes for the aerosol to form could thus affect the representation of the cloud. The nine days used here is shorter than previous estimates of volcanic $SO_2$ conversion 10 in the stratosphere (Andersson et al., 2013), but ongoing $SO_2$ conversion adds uncertainty to our estimation. The short time span covered in the present study can thus be assumed to have a small effect on $\beta_{aer}$ per $SO_2$. This can be seen in Fig. 8 in Friberg et al. (2018), where the stratospheric aerosol load from Sarychev peaks in September, i.e. months after the last swath used in this study.

15 In Fig. 6 we show a snapshot of the FLEXPART transported aerosol scattering from all 75 CALIOP swaths at the times of the AIRS swaths in Fig. 2 (similar to the grey areas shown in the concept sketch Fig. 1). The geographical extents have been chosen to focus on the regions of Fig. 2 with the most $SO_2$ for each of the five orbits. Starting from the rightmost subfigure and earliest time; the transported aerosol scattering outlines the largest $SO_2$ cloud seen by AIRS. At midnight to June 19[th] the more southern clouds are outlined, while the northern string of $SO_2$ is sparsely covered by time-adjacent CALIOP swaths. At 20 01:30 UTC, the transported aerosol scattering fails to outline the faint $SO_2$ cloud. In the two final comparisons, the transported aerosol scattering is centred on the $SO_2$ but also have difficulties in contouring the more peripheral $SO_2$. While the transported aerosol scattering generally outline the $SO_2$ seen with AIRS, there are a few mismatches and false positives of where the FLEXPART transported aerosol scattering indicate $SO_2$ presence where there is none detected by AIRS.

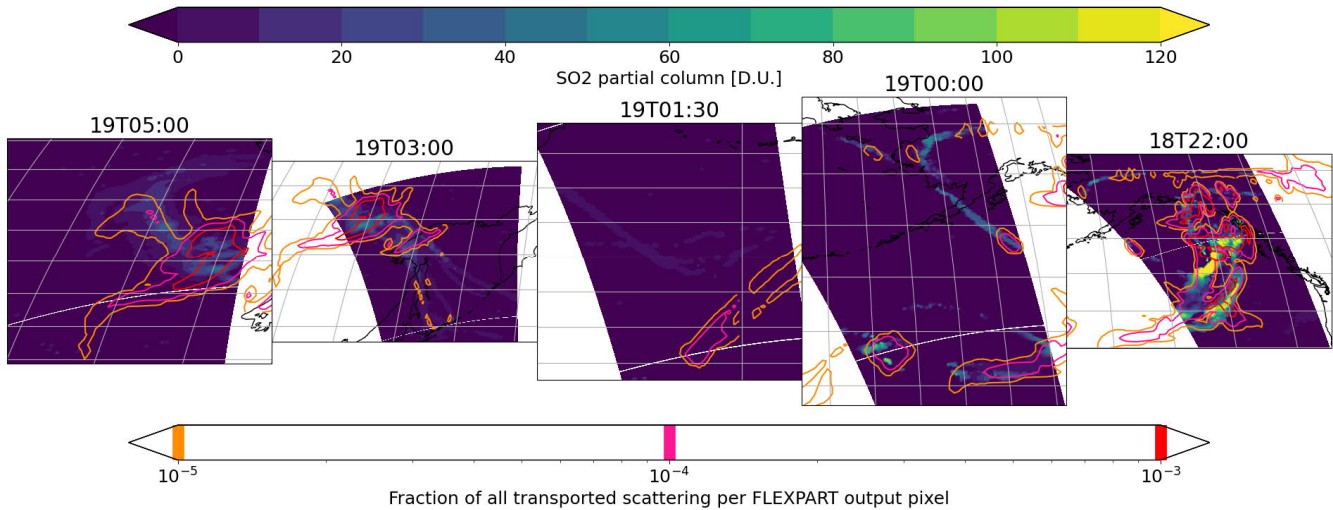

**Figure 6: Overview of the transport of vertical profiles from the CALIOP swaths to the time of the AIRS swaths from Fig. 2. The figures are zoomed to the areas of the most SO$_2$ for ease of comparison. To get a flat horizontal mapping of the transported CALIOP swaths, the combined profiles were integrated vertically. Percentages of SO$_2$ mass outside the $10^{-5}$ contour compared to the total SO$_2$ mass are: 0.5 %(18T22:00), 0.8 % (19T00:00), 0.7 % (19T01:30), 0.7 % (19T03:00), and 0.6 % (19T05:00).**

The overlap between the transported aerosol and the SO$_2$ from Fig. 2 varies for the five orbits of the AIRS instrument, which is shown in Fig. 6 where the rightmost subfigure contains the first orbit. While the aerosol generally outlines the horizontal location of SO$_2$ during the first, fourth and fifth orbits, the transported aerosol does not contain the SO$_2$ seen during the second and third orbits. For the second orbit at midnight in Fig. 6, the northern part of the SO$_2$ is scarcely covered by the transported aerosol with 0.0086 Tg (0.8 % of all SO$_2$ in the AIRS swaths) outside the $10^{-5}$ contour. The reason for this could be that the SO$_2$ is located below the tropopause, but the aerosol scattering in the closest CALIOP swath is located only in the stratosphere. Also, CALIOP does not observe all parts of the volcanic cloud and at some locations the height information can come from a relatively weak CALIOP profile. Therefore, we chose to horizontally average the combined transported aerosol within each individual AIRS swath. This horizontal averaging was done by first interpolating the combined transported CALIOP data from FLEXPART to the coordinates of the AIRS pixels. Then a horizontal averaging was made over the whole AIRS footprint to get a representative height profile. This averaging accounts for when FLEXPART and the observed SO$_2$ are slightly mismatched or when there is limited coverage from the transported CALIOP swaths. The averaging created a single height distribution for each AIRS swath containing observations of the volcanic clouds. The horizontal averaging also compensates for the fact that CALIOP has limited spatial coverage which means that not all parts of the AIRS cloud was completely covered by CALIOP's height information transported by FLEXPART. Of the 75 CALIOP swaths, 69 have trajectories that enter the main SO$_2$ AIRS observations. The 6 CALIOP swaths whose trajectories do not enter the AIRS observations in Fig. 2 contain little aerosol scattering. There is a list of which CALIOP swaths are transported into which AIRS swaths in the supplementary material.

## 3.4 Collecting height resolved AIRS swaths into a complete collection

The collection of AIRS swaths from Fig. 2 was found to cover almost all of the volcanic clouds from Sarychev during several hours of orbit, see Fig. 7a. One AIRS swath takes 6 minutes to scan. Thus, there are 240 swaths for each day. In this study, AIRS swaths will be referenced as "<day of June 2009>.<swath number of day>", e.g. 18.223 means the 223rd swath at June

18th. The height profiles for the AIRS swaths in this collection are shown in Fig. 7b-d. Height distributions using geometric altitude and pressure as height coordinates were calculated from the potential temperature using ERA5 temperature data for the AIRS footprints. The height profiles in Fig. 7b-d clearly show that the clouds over the eastern Pacific Ocean (130°W) are located at higher altitudes than the clouds over eastern Siberia (130°E).

From this collection of AIRS swaths, a 3D global dataset was compiled for implementation in models. The dataset has 1° latitude × 1° longitude horizontal resolution and comes in the three different versions depending on which vertical coordinates that is preferred (potential temperature, geometric altitude, and pressure). The vertical resolutions are 1 K for the potential temperature grid, 200 m for geometric altitude, and for the pressure grid the pressure levels correspond to geometric altitude steps of 200 m. The dataset will be made available through www.snd.gu.se (when the manuscript has been accepted).

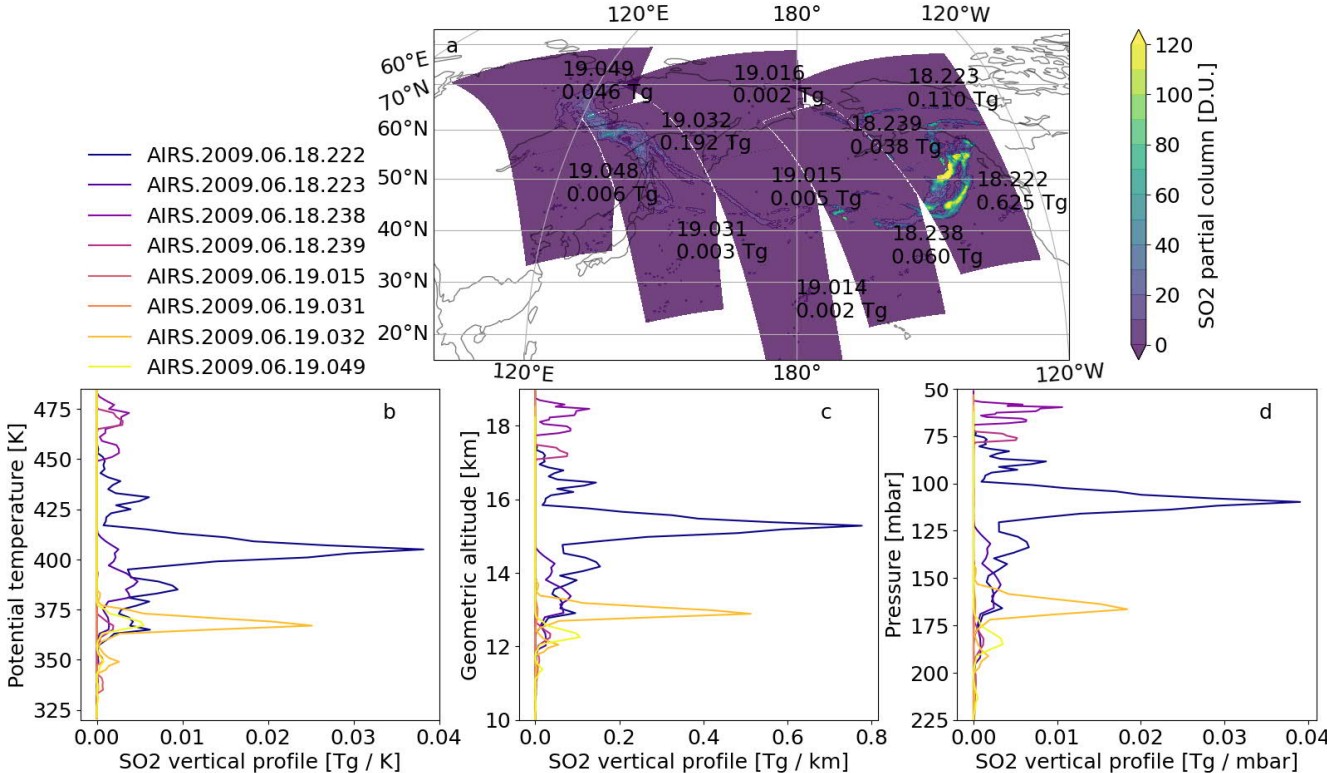

**Figure 7: a) A collection of AIRS SO$_2$ swaths spanning the entire volcanic cloud. The SO$_2$ mass in the wedge of missing coverage between swaths 18.222, 18.238, 18.239 is estimated to be 0.036 Tg (3 % of the total mass). b-d) Height profiles for the AIRS swaths in the collection shown in a), in potential temperature (b), geometric altitude (c), and pressure (d). The number of CALIOP swaths that had air tracers transported into the SO$_2$-swaths were: 18.222 (34 CALIOP swaths), 18.223 (34), 18.238 (17), 18.239 (25), 19.014**

**(1), 19.015 (5), 19.016 (6), 19.031 (5), 19.032 (20), 19.048 (8), and 19.049 (15). Vertical profiles were not compiled for three of the swaths in a), since the volcanic SO2 concentration was very low and no CALIOP data could be matched with these AIRS swaths.**

## 4 Discussion

The clouds over the eastern Pacific Ocean have broad vertical distributions (Fig. 7b-d), probably due to a higher number of

5  eruption emissions being transported into this region and their variation in injection height. In Fig. 8, we show a comparison between the sum of our height profiles and the height distributions that other studies have reported on or used as input in model simulations. We sum profiles at swaths during the 18[th] and 19[th] of June whereas in three modelling studies (Haywood et al., 2010;Mills et al., 2016;Lurton et al., 2018) sulphur is released closer in time to the eruption. We have no information about the potential temperature at the release points in the other studies, so they are compared using geometric altitude.

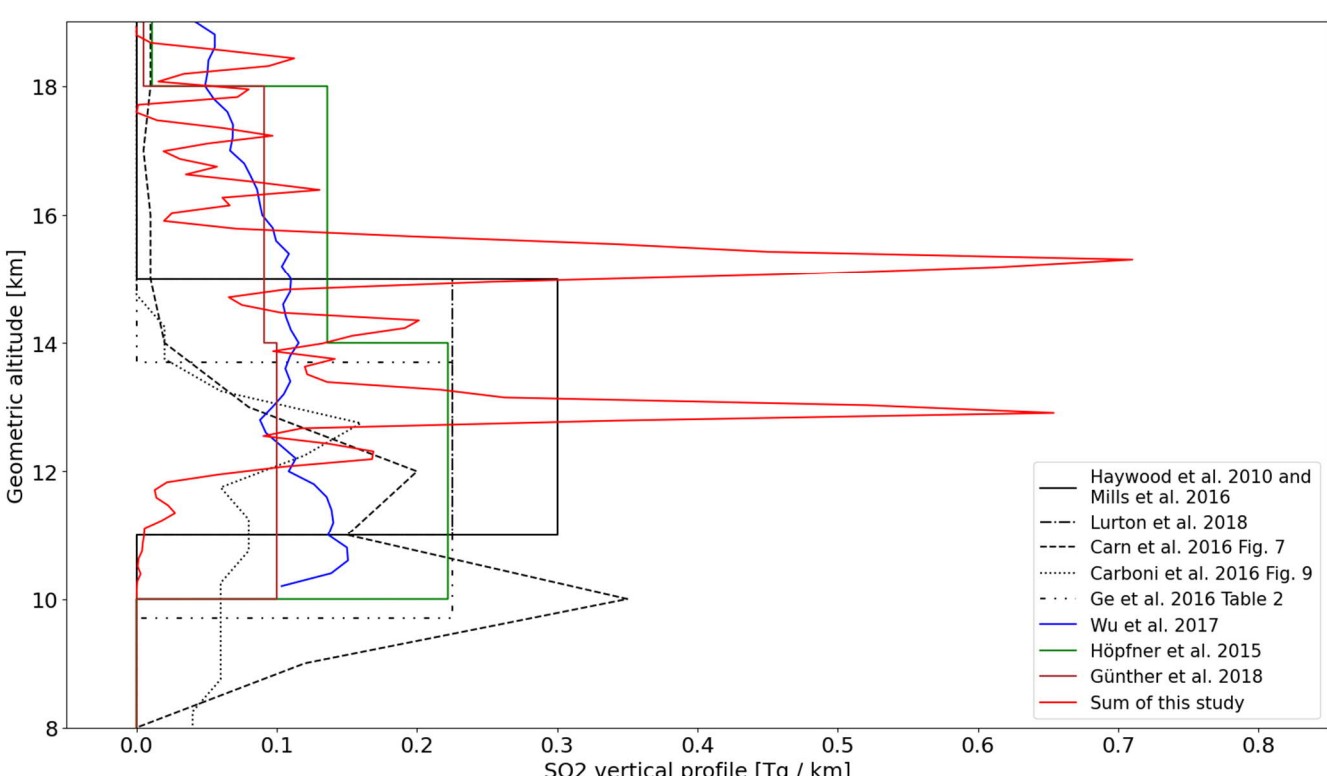

**Figure 8: SO2 profiles from other studies compared with the sum of our profiles from Fig. 7c. The emissions in the model studies**
15  **were released on: Haywood et al. 2010 - 2009-06-15 – 2009-06-16, Lurton et al. 2018 – 2009-06-15, Ge et al. 2016 – 2009-06-11, Günter el at. 2018 – 2009-06-12, Höpfner et al. 2015 – 2009-06-12.  For Carn et al. 2016 and Carboni et al. 2016 the data closest in time to our dataset was used. In Wu et al. 2017 SO2 emissions are integrated between the 2009-06-12 and 2009-06-16.**

A majority of the other studies have all of the $SO_2$ mass placed below 15 km, while roughly half of the $SO_2$ reported by us is above this limit. This altitude coincides with the potential temperature level of 380 K in the release area of this study, which is the upper limit of the LMS. The residence time and transport paths for sulphate aerosol is substantially different in the LMS compared to the overlying stratosphere. Therefore, placing the entire $SO_2$ mass from the eruption in the LMS instead of half above it should have impacts on the aerosol residence time and climate impact when using these data in modelling studies.

The Sarychev $SO_2$ profiles by Wu et al. (2017), Höpfner et al. (2015), and Gunther et al. (2018) are more similar to our profiles in terms of the mass distributed at higher altitudes. Nevertheless, their profiles do not contain the two strong peaks that is present in our profile at 13 and 15 km altitude. The peak above 15 km is associated with the $SO_2$ cloud located over the Eastern Pacific while the one at 13 km is associated with the $SO_2$ cloud located over Eastern Siberia. As can be seen in Fig. 5 the volcanic clouds often have a limited vertical extent. Since we use the altitude information from CALIOP with its high resolution, our dataset captures the limited vertical extent of the volcanic clouds, resulting in peaks in the overall profile. The peaks do not appear in the other datasets as their methods are likely not capable of resolving the heights of the individual sub-clouds. That our dataset can resolve the variation in height between the different sub-clouds enables more realistic modelling of the movement of the Sarychev volcanic clouds as wind profiles can vary strongly with geographical location.

More than half of the $SO_2$ in Fig. 7b is above the 380 K isentrope. This is in agreement with the distribution in Fig. 8 in the long-term aerosol load study of Friberg et al. (2018), where July data shows that more than half the stratospheric aerosol optical depth was located above the 380 K isentrope. This means that a large fraction of the aerosol is in layers that are isentropically connected to the tropical stratosphere, as Fig. 7 in Friberg et al. (2018) also shows. Thus, the results from the method presented in this paper, based on early observations after the eruptions, agree with observations at the time when most of the aerosol has formed from the volcanic $SO_2$ emissions. This shows that the salient features of the vertical distribution of the volcanic clouds are caught by our method.

While our method fails to ascribe height distributions to individual pixels in an AIRS swath, it provides height distributions with high vertical resolution representative of entire AIRS swaths. Note that our method is not dependent on which $SO_2$ instrument is used. The core of the method is to make use of many CALIOP observations and detailed FLEXPART simulations to transport the vertical information in CALIOP swaths to the time and place of other less vertically resolved measurements. Our approach works well for Sarychev and will likely do so for other eruptions. However, for eruptions with large amounts of ash one might need to check the impact from sedimentation of ash using the depolarization and colour ratios in the CALIOP data.

Our method could also be simplified further by not resolving the height distributions of individual sub-clouds, and instead resolve only the emitted $SO_2$ as a whole. This simplified method would then start with finding stratospheric clouds in CALIOP

swaths. Ensuring that they do not contain heterogeneous ash distributions would be done by checking the depolarization ratios. Then a $SO_2$-satellite instrument would verify that the aerosol cloud obtained from CALIOP is of volcanic origin. Finally, the approved CALIOP clouds could all be weighted together, keeping the potential temperature level at the original measurement positions, by relating to the scattering intensity from CALIOP in each pixel. This simplified approach would not need

FLEXPART, and would be much more accessible and rapid.

## 5 Conclusions

We have shown how a large number of lidar observations of fresh volcanic aerosol particles can provide a vertical dimension to passive horizontal sulphur dioxide observations, which would improve volcanic inputs to climate models. Lidar measurements cover narrow stretches of the atmosphere and are on their own difficult to link to the wider observations by

passive instruments. To remedy this, CALIOP data were prepared in a semi-automatic way to be used as input to FLEXPART. To ensure a smooth interface between CALIOP and FLEXPART, the fine resolution of the aerosol layers in the CALIOP data were preserved when entered into FLEXPART as air tracers. Volcanic sulphate aerosol particles, which CALIOP observe light scattering from, are created out of the emitted $SO_2$ gas, which is observed by passive satellite instruments. In the method, we use the high vertical resolution of CALIOP, assuming that the particles and the $SO_2$ are co-located.

Previously published $SO_2$ vertical distributions used for modelling purposes have a lower vertical resolution and oftentimes places the $SO_2$ from the Sarychev eruption at too low altitude compared with our results. Our deduced vertical $SO_2$ distribution from the first two weeks after the eruptions shows good agreement with published vertical high-resolution aerosol profiles describing the conditions during the weeks and several months after the eruption, i.e. when almost all $SO_2$ is converted to

sulphate particles. Here we have demonstrated the method for one volcanic eruption (Sarychev 2009). We find that this method increases the vertical resolution and attainable accuracy compared to previous studies of the $SO_2$ vertical profiles in the stratosphere following the studied volcanic eruptions.

### Acknowledgement

We would like to thank Fred Prata for providing us with the AIRS $SO_2$ data and for patiently answering our questions. We are

grateful for the thorough reviews by Marc von Hobe and one anonymous referee that helped us improve the manuscript. We also thank Xue Wu, Sabine Griessbach, and Lars Hoffmann for their discussion comment and for providing us with $SO_2$ profile data. The CALIOP data (NASA/LARC/SD/ASDC, 2016) comes from the joint satellite mission between NASA and CNES.

**Financial support**

This research was supported by the Swedish National Space Agency (contracts 130/15, 104/17), FORMAS (contracts 2018-00973 and 2020-00997), and The Crafoord Foundation (contract 20190690).

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
