# Peer review of "Methodology to obtain highly resolved SO2 vertical profiles for representation of volcanic emissions in climate models"

_Atmospheric Measurement Techniques, 2021_

## Referee Comment (RC1)

Review of "*Methodology to obtain highly resolved SO$_2$ vertical profiles for representation of volcanic emissions in climate models*" by Oscar S. Sandvik and co-authors

The manuscript describes a method employing trajectory calculations to combine information from two satellite instruments and deduce the vertical distribution of SO$_2$ in fresh volcanic plumes. Such a multi-sensor analysis method that involves some modelling is probably at the edge of the scope of Atmospheric Measurement Techniques, but because the output can be regarded as higher level observational data, I would still rate it within the journal's scope. The manuscript is appropriately structured and well written and the figures are well composed and of high quality.

The one thing that I find not (yet) convincing is the value and purpose of the dataset created by the new method. And there are a few aspects of the method that are not fully clear to me and that may bear some caveats, which in my opinion are not properly discussed. I will touch on these issues in some general points below and then point out a few more specific points that may need attention but that are probably minor in nature.

**General points:**

1. At the end of the abstract, a "gridded high vertical resolution SO$_2$ inventory that can be used in Earth system models" is mentioned. The retrieval of the SO$_2$ height distribution in AIRS swaths is well described, but I'm not sure I would classify that as "gridded inventory". Of course, for global Earth system models, SO$_2$ data on a regular horizontal/vertical/temporal grid would be of great value. If your data product is anything like such a larger gridded inventory, then please do describe this in more detail (on page 10, you describe gridding the FLEXPART output on a 0.5 x 0.5 degree grid, but that is prior to linking it to the AIRS information, so it is not your final "product", or is it?).

2. In the same context, the simpler (compared to a "gridded inventory") vertical information for volcanic plumes/clouds/sub-clouds may still be very useful for modellers. But I wonder if hinging them to the AIRS swaths in space and time is the most useful thing that can be done. If you assume that the vertical distribution of the aerosols observed by CALIOP is representative for the SO$_2$ distribution, then why not calculate everything right back to the volcano? Vertically resolved emission plumes could then be "injected" into models of any scale.

3. There has been a rather comprehensive study on the Sarychev plume and its dispersion by Wu et al. (2017), and I was rather surprised to see no mention of it in your paper!

   I suggest you look carefully at this previous work that also combines satellite observations and trajectory calculations. It should probably be mentioned in your introduction, and a comparison of the two studies (What was done similar? What different? Are the results in agreement?) in your discussion is clearly warranted.

   Note that while I have been working on this review, a short comment was uploaded by the authors of the Wu et al. (2017) paper. This comment contains some excellent ideas and suggestions as to how connections between the two studies could be made, and I strongly recommend you to carefully look at this short comment and follow the recommendations.

4. According to your method description, you use ERA-5 reanalysis data to drive the FLEXPART trajectory model and MERRA-2 to determine tropopause heights and potential temperature levels of the CALIOP data (Page 7, lines 22 – 25). Would it not be more consistent to use one single reanalysis data set for both?

5. Can you give more detail on how horizontal transport and transforms between different vertical coordinates is realized? On page 9, lines 6 – 10, you explain that potential temperature "is stored" for the released tracer particles. On page 10, lines 24 – 26, you state that you "did not use the FLEXPART output vertical coordinates" and "set a single height interval between 0 and 50 km in the output grid specification". I find this confusing. Do you mean that you transport the parcels from the CALIOP SWATH converting geometric altitude to potential temperature in the beginning and then convert the same potential temperature value back to geometric altitude at the end? And when you say "locations" on page 9, line 10, does that include time in that case?

    If the potential temperature of an air parcel is indeed not allowed to change during transport in your simulations, then the method would obviously not work in the tropics, where cross-isentropic vertical transport is common. Also, self-lofting in fresh volcanic plumes would not be accounted for, correct?

    Some clarifications and a more extensive discussion of this would help.

6. I do wonder to what extent the issues and uncertainties I mentioned under points 4 and 5 above could factor in producing the bimodal distribution in the sum of your study that is shown in Figure 7, and also on the height of your $SO_2$ plume being somewhat higher than the earlier studies you mention. First, the relationship between potential temperature and geometric altitude is probably not the same at the different locations (I guess the profiles from all the earlier studies are close to the volcano in space and time?), so when you strictly transport along isentropes, it may be better to do this comparison on a potential temperature vertical coordinate. Second, self-lofting could explain at least part of the difference in altitude and should be mentioned.

7. As you state on page 3, lines 8/9, an important assumption for your method to work is the horizontal and vertical co-location of volcanic $SO_2$ and aerosol following the eruption. While this assumption is probably reasonable for a wide range of eruptions and conditions, I still suggest a more detailed discussion to further support this assumption: what do we know from the literature with respect to the $SO_2$/aerosol co-location? Are there conditions, under which the assumption may not hold, and on what time scales will it hold? How about ash-rich eruptions, where particle sedimentation rates may be higher (you mention this at the end of page 6)?

    This may all not be critical for your particular case looking at rather short time scales not so long after the eruption for a non-tropical eruption. But as you describe a new method that should hopefully be applicable to other cases, such caveats and limitations should be made more clear.

**Specific and minor points:**

Page 2, line 15: The Watts (2000) reference for OCS sources and sinks is in the middle of the timeline of possible references, i.e. it is neither the first one nor is it up to date. I suggest to cite Kremser et al. (2016, already in your reference list), which contains a fairly recent OCS budget together with a discussion of source and sink processes.

Page 9, lines 1 – 5: I'm not sure that I have fully understood the scaling of the number of FLEXPART particles. Are you saying that you are releasing $x$ FLEXPART particles per one CALIOP pixel, and $x$ scales with $\beta_{aer}$ of that pixel?

Page 12, lines 2 – 3: As the reason for the lack of trajectories matching some of the AIRS $SO_2$ clouds, you mentioned that these $SO_2$ detections could be located below the tropopause, while only the stratospheric aerosol scattering observed by CALIOP is "transported". Have you tried testing this by, e.g., using TP - 2km instead of TP as vertical threshold?

**References:**

Wu, X., Griessbach, S., and Hoffmann, L.: Equatorward dispersion of a high-latitude volcanic plume and its relation to the Asian summer monsoon: a case study of the Sarychev eruption in 2009, Atmos. Chem. Phys., 17, 13439–13455, https://doi.org/10.5194/acp-17-13439-2017, 2017.

---

## Community Comment (CC1)

Dear authors,

Thank you for sharing your interesting work. We would like to point you to our study dating back to 2017 (Wu et al., 2017) because we think it provides a good comparison to your results. In Wu et al. (2017), we focused on the Sarychev eruption in 2009 using AIRS $SO_2$ observations and a Lagrangian particle dispersion model to reconstruct the altitude-resolved $SO_2$ emission time series. The following figure (Fig. 2 in Wu et al., 2017) shows the $SO_2$ emission time series of Sarychev during its entire eruption time period.

[Figure]

**Figure 2.** Sarychev $SO_2$ emission time series derived from AIRS measurements using a backward trajectory approach (see text for details). The emission data are binned every 1 h and 0.2 km. Black dots denote the height of the thermal tropopause.

We verified the reconstructed $SO_2$ emission time series by using it to initialize forward trajectories and compared the simulated horizontal dispersion with further AIRS $SO_2$ and MIPAS aerosol observations. In the vertical, the comparison with altitude resolved MIPAS aerosol measurements showed good agreement.

In addition to the Sarychev case, the method was also applied to other volcanic eruption cases (Grímsvötn, Nabro, Puyehue-Cordón Caulle, and Merapi) in Hoffmann et al. (2016) and Wu et al. (2018). With a more computationally expensive inverse approach, we investigated the Nabro eruption in Heng et al. (2016).

Since we used a very similar methodology and applied it to the same eruption case, we would appreciate it if you would spend some time reading our published work, particularly Wu et al. (2017) and Hoffmann et al. (2016), and perhaps compare your results with ours.

For your convenience, I calculated the Sarychev $SO_2$ mass profile on 14 and 15 June 2009 (by adding up all $SO_2$ mass on each level for each day) as below:

[Figure]

You may see the two peaks in the vertical direction at 10–12 km and 14–16 km, which seems to agree with your results as shown in your figure (although the date differs). If you are interested in comparing your results with ours, please do not hesitate to ask Dr. Hoffmann (l.hoffmann@fz-juelich.de) or me (wuxue@mail.iap.ac.cn) for our data.

Thanks again for your interesting work.

Sincerely,
Lars, Sabine, and Xue

References:

Hoffmann, L., Rößler, T., Griessbach, S., Heng, Y., and Stein, O.: Lagrangian transport simulations of volcanic sulfur dioxide emissions: Impact of meteorological data products, J. Geophys. Res. Atmos., 121, 4651– 4673, doi:10.1002/2015JD023749, 2016.

Heng, Y., Hoffmann, L., Griessbach, S., Rößler, T., and Stein, O.: Inverse transport modeling of volcanic sulfur dioxide emissions using large-scale simulations, Geosci. Model Dev., 9, 1627–1645, https://doi.org/10.5194/gmd-9-1627-2016, 2016.

Wu, X., Griessbach, S., and Hoffmann, L.: Equatorward dispersion of a high-latitude volcanic plume and its relation to the Asian summer monsoon: a case study of the Sarychev eruption in 2009, Atmos. Chem. Phys., 17, 13439–13455, https://doi.org/10.5194/acp-17-13439-2017, 2017.

Wu, X., Griessbach, S., and Hoffmann, L.: Long-range transport of volcanic aerosol from the 2010 Merapi tropical eruption to Antarctica, Atmos. Chem. Phys., 18, 15859–15877, https://doi.org/10.5194/acp-18-15859-2018, 2018.

---

## Author Comment (AC1)

We thank Lars, Sabine, and Xue for reaching out to us.

We were not aware of the Wu et al. paper when writing the manuscript, and are relieved that the authors contacted us at such an early stage. We have contacted the authors and added their data to Fig. 7 (now Fig. 8). We added their study in the introductions section and added a reference to their study.

---

## Author Comment (AC2)

We thank the reviewers for constructive feedback that has improved our manuscript. The reviewer comments are given below in black font followed by our answers in this blue color.

General points:

1. At the end of the abstract, a "gridded high vertical resolution SO2 inventory that can be used in Earth system models" is mentioned. The retrieval of the SO2 height distribution in AIRS swaths is well described, but I'm not sure I would classify that as "gridded inventory". Of course, for global Earth system models, SO2 data on a regular horizontal/vertical/temporal grid would be of great value. If your data product is anything like such a larger gridded inventory, then please do describe this in more detail (on page 10, you describe gridding the FLEXPART output on a 0.5 x 0.5 degree grid, but that is prior to linking it to the AIRS information, so it is not your final "product", or is it?).

We agree with the reviewer. We have compiled a 3D-gridded products and added a paragraph on the 3D-dataset in the results section. "...The dataset has 1° latitude × 1° longitude horizontal resolution and comes in the three different versions depending on which vertical coordinates that is preferred (potential temperature, geometric altitude, and pressure). The vertical resolutions are 1 K for the potential temperature grid, 200 m for geometric altitude, and for the pressure grid the pressure levels correspond to geometric altitude steps of 200 m. The dataset is available as a supplementary file..."

2. In the same context, the simpler (compared to a "gridded inventory") vertical information for volcanic plumes/clouds/sub-clouds may still be very useful for modellers. But I wonder if hinging them to the AIRS swaths in space and time is the most useful thing that can be done. If you assume that the vertical distribution of the aerosols observed by CALIOP is representative for the SO2 distribution, then why not calculate everything right back to the volcano? Vertically resolved emission plumes could then be "injected" into models of any scale.

We agree that it could be useful to use only few vertical profiles (one per eruption) at the geographical location and time of eruptions. However, the aim of our study is to provide SO2 profiles by combining the different strengths of the different satellite instruments. Our data-set can be implemented directly into models, and would likely be at least as good as injection profiles from the volcanoes site, maybe better: Starting with a single (or few) horizontal grid point(s), small errors in simulated transport paths in the first few days may lead to larger errors than starting out with a SO2 distribution already spread over a larger area. This is particularly useful for Earth system models with coarser resolution that may have limited ability to capture the initial transport.

3. There has been a rather comprehensive study on the Sarychev plume and its dispersion by Wu et al. (2017), and I was rather surprised to see no mention of it in your paper! I suggest you look carefully at this previous work that also combines satellite observations and trajectory calculations. It should probably be mentioned in your introduction, and a comparison of the two studies (What was done similar? What different? Are the results in agreement?) in your discussion is clearly warranted. Note that while I have been working on this review, a short comment was uploaded by the authors of the Wu et al. (2017) paper. This comment contains some excellent ideas and suggestions as to how connections between the two studies could be made, and I strongly recommend you to carefully look at this short comment and follow the recommendations.

We were not aware of the Wu et al. paper when writing the manuscript, and are relieved that the authors contacted us at such an early stage. We have contacted the authors and added their data to Fig. 7 (now Fig. 8). We added their study in the introductions section and added a reference to their study.

4. According to your method description, you use ERA-5 reanalysis data to drive the FLEXPART trajectory model and MERRA-2 to determine tropopause heights and potential temperature levels of the CALIOP data (Page 7, lines 22 – 25). Would it not be more consistent to use one single reanalysis data set for both?

We agree that it is generally preferable to use the same reanalysis data. However, FLEXPART uses the ERA-5 data, whereas the CALIOP hdf-files contain meteorological data from MERRA-2 simulated to the specific locations and time of the sensor's observations.

5. Can you give more detail on how horizontal transport and transforms between different vertical coordinates is realized? On page 9, lines 6 – 10, you explain that potential temperature "is stored" for the released tracer particles. On page 10, lines 24 – 26, you state that you "did not use the FLEXPART output vertical coordinates" and "set a single height interval between 0 and 50 km in the output grid specification". I find this confusing. Do you mean that you transport the parcels from the CALIOP SWATH converting geometric altitude to potential temperature in the beginning and then convert the same potential temperature value back to geometric altitude at the end? And when you say "locations" on page 9, line 10, does that include time in that case? Yes, it includes time. We see that this may be confusing. Thank you for pointing this out. We have made adjustments in the manuscript to clarify how this procedure works (at the end of section 2.2). The geometric altitude is first transformed to potential temperature, computational particles are then transported, and finally we transform back to geometric altitude. (The first step is to convert geometric altitudes to potential temperatures. Each computational particle gets a potential temperature value that follows the particle during its transport. After transport, the particles are put on a potential temperature grid to sum up the amount of particles in each potential temperature bin.).

If the potential temperature of an air parcel is indeed not allowed to change during transport in your simulations, then the method would obviously not work in the tropics, where cross-isentropic vertical transport is common. Also, self-lofting in fresh volcanic plumes would not be accounted for, correct? Some clarifications and a more extensive discussion of this would help.

It is true that our method cannot capture diabatic transport, but this should be a minor issue due to the short time span needed for the analysis. It is important to use potential temperature to follow the aerosol transport in the stratosphere. Dense aerosol clouds of supermicron particles can lead to radiative heating and self-lofting, but this effect should have little impact on the smaller particles from the Sarychev eruptions. We agree that it would be difficult to use this approach for a situation where we have rapid radiative heating of a dense aerosol layer with supermicron particles like dense wildfire smoke. This may be the case also for Mt Pinatubo's and larger eruptions' volcanic clouds. For the present eruption, similar-sized, and smaller eruptions, the potential temperature should be contained during the short time-frame in our study (few days backwards and forwards). It is worth noting that Fairlie et al., (2014) found a self-lofting of no more than 0.3 K/day for aerosol from the

June 12, 2011 eruption of Nabro. Nabro injected similar amounts of SO2 as Sarychev did. Hence, self-lofting should not have any major impact on our SO2 profiles.

We added a statement on self-lofting and its impact on our data in Section 3.1.

6. I do wonder to what extent the issues and uncertainties I mentioned under points 4 and 5 above could factor in producing the bimodal distribution in the sum of your study that is shown in Figure 7, and also on the height of your SO2 plume being somewhat higher than the earlier studies you mention. First, the relationship between potential temperature and geometric altitude is probably not the same at the different locations (I guess the profiles from all the earlier studies are close to the volcano in space and time?), so when you strictly transport along isentropes, it may be better to do this comparison on a potential temperature vertical coordinate. Second, self-lofting could explain at least part of the difference in altitude and should be mentioned.

The vertical distribution in our figure is in agreement with CALIOP observations both in July and few months after the eruptions of Sarychev (Fig. 8 in Friberg et al. 2018). The bimodal structure has to do with multiple eruptions from the volcano over the course of a few days. This is indicated in Fig. 6 in the manuscript where we see different vertical profiles at different locations. The Eastern parts are located at ~3 km (40 K) lower altitudes than the Western. With the numbers given by Fairlie et al. (2014), the lofting in our study period (9 days) would be 2.7 K ( ~160 m). Note that this would be the maximum possible lofting according to their study, i.e. much smaller than the height difference between the different sub-clouds. Given these numbers, self-lofting cannot explain the bimodality in Fig. 7. More details regarding self-lofting are discussed in our answer to comment #5 above.

The potential temperature was converted back to geometric altitude at the locations of the AIRS pixels. Hence, there should be no mismatch from our approach. We agree that it would be preferable to compare all studies on potential temperature coordinates, but most studies only report geometric altitudes or pressure levels. Thus, we preferred to transform our dataset to the individual AIRS swaths' locations instead of assuming a climatological transform function for the published data that we compared our data to.

7. As you state on page 3, lines 8/9, an important assumption for your method to work is the horizontal and vertical co-location of volcanic SO2 and aerosol following the eruption. While this assumption is probably reasonable for a wide range of eruptions and conditions, I still suggest a more detailed discussion to further support this assumption: what do we know from the literature with respect to the SO2/aerosol co-location? Are there conditions, under which the assumption may not hold, and on what time scales will it hold? How about ash-rich eruptions, where particle sedimentation rates may be higher (you mention this at the end of page 6)? This may all not be critical for your particular case looking at rather short time scales not so long after the eruption for a non-tropical eruption. But as you describe a new method that should hopefully be applicable to other cases, such caveats and limitations should be made more clear.

We agree that it is important to consider possible ash signals. CALIOP retrieves a depolarization ratio that we use as a means of checking the impact of ash. We investigated the homogeneity of the depolarization ratios' vertical distributions and find no evidence of strong sedimentation during the studied period (section 2.2). The fact that the vertical distribution in the present study matches CALIOP observations in weeks and months later is evidence for that ash has little impact on our retrieved SO2 profiles. We agree with the reviewer that it is valuable to clarify that it may differ

depending on the eruption and added text to the manuscript clarifying that this may not always be the case, but that it worked fine for Sarychev (Section 2.2).

Specific and minor points:

Page 2, line 15: The Watts (2000) reference for OCS sources and sinks is in the middle of the timeline of possible references, i.e. it is neither the first one nor is it up to date. I suggest to cite Kremser et al. (2016, already in your reference list), which contains a fairly recent OCS budget together with a discussion of source and sink processes. Thank you for pointing this out. We now refer to Kremser instead of Watts.

Page 9, lines 1 – 5: I'm not sure that I have fully understood the scaling of the number of FLEXPART particles. Are you saying that you are releasing x FLEXPART particles per one CALIOP pixel, and x scales with $\beta$aer of that pixel? Yes, the released number of particles scales with the strength of the CALIOP signal (light backscattering). This enable us to run CALIOP swaths with both strong and weak backscattering signals. We changed the text to clarify (Sections 3.2 and 3.3).

Page 12, lines 2 – 3: As the reason for the lack of trajectories matching some of the AIRS SO2 clouds, you mentioned that these SO2 detections could be located below the tropopause, while only the stratospheric aerosol scattering observed by CALIOP is "transported". Have you tried testing this by, e.g., using TP - 2km instead of TP as vertical threshold? We have checked the occurrence of volcanic aerosol in the vicinity of the tropopause. It is difficult to separate ice-clouds from volcanic aerosol in the troposphere, and more so the further away from the tropopause. We therefore cannot include the troposphere in our analysis.

---

## Author Comment (AC4)

We thank the reviewers for constructive feedback that has improved our manuscript. The reviewer comments are given below in black font followed by our answers in this blue color.

This paper presents a new method deriving vertical profiles of SO2 representative of volcanic emissions by combining space-borne observations of SO2 by AIRS/Aqua and of aerosol by CALIOP/Calipso. The profiles are constructed by connecting the CALIOP high resolution aerosol profiles with the AIRS SO2 columns using trajectory calculations. The interest of providing SO2 profiles at the period of a volcanic injection in the stratosphere is definitely clear for the community especially for the initialization of Chemistry-Transport and Climate Models that aim at estimating the chemical and radiative impacts of volcanoes. So I found the idea behind the manuscript of very high scientific return. I found the manuscript structure pertinent, well-written and going straight to the obtained results (but a bit too much). Although I estimate that this work is really worthy of publication in AMT, there are some methodological elements that still need to be clarified throughout the text.

General comments:

It can be difficult to grasp how the trajectories interconnect both satellite datasets (e.g. how they intersect, typical trajectory lengths). An illustration of a specific case linking CALIOP at a given altitude level and a AIRS swath would be helpful for reader. The relevance of using forward trajectories starting from CALIOP aerosol profiles (aerosol are a product of SO2 so I much better clearly understand the use of backward trajectories) can be clarified too . Why in the example of the command file the simulation lasts for 6 days?

We ran all FLEXPART simulations for the same time-span (June 14th – 23rd) for programming simplicity. (Forward trajectories were run from the CALIOP sampling time to the 23rd June, and backward trajectories were run from the CALIOP sampling time to the 14th). However, we only used the parts of the trajectories needed to connect the CALIOP profiles to the AIRS swath of interest.

The reason for using both forward and backward trajectories is that it enables us to use the information from as many CALIOP swaths as possible for each AIRS swath. Height information from each CALIOP swath was transported by FLEXPART to the AIRS (at their specific time and location of the AIRS swaths). Also using forward trajectories enabled us to use more CALIOP swaths, and hence more height information, in our analysis. In order to make this clearer we have included a conceptual figure of the method in the introduction (Fig. 1 in the new version of the manuscript).

To me (but I may be mistaken!) the authors should find CALIOP and AIRS swaths capturing the volcanic plume every day, allowing them to compute trajectories of a few hours only to connect both satellite observations.

There are CALIOP and AIRS swaths every day, however only using CALIOP data from the same day as the AIRS swath limits our analysis to a very small part of the volcanic cloud since the swath width of CALIOP is very narrow. We include information from CALIOP swaths from several days to have better height information from a larger part of the volcanic cloud.

We understand that this was not properly addressed and have changed the text (in the abstract as well as in the introduction) to clarify the advantage of using multiple CALIOP swaths.

A clearer description of the limitation in the method is missing. For instance, since what would be the impact of the sedimentation of the aerosols in the time length of the trajectories that on the attribution of the retrieved SO2 plume heights? I guess the impact would be limited but I would suggest the authors to mention it.

Yes, the sedimentation rates for fine mode particles are negligible in the time-frame and vertical resolution of our study. We agree that this is worth mentioning and added text to the manuscript (section 2.2).

In case of the presence of ash which has been reported for some eruptions (Kelud, Raikoke) and as derived from CALIOP depolarization/colour ratios, how the method would be affected? Can the CALIOP instrument adequately distinguish between ash and sulfate to properly associate the aerosol detection with the corresponding SO2 observation away from it?

As the reviewer suggests, the sedimentation of ash would be spotted by CALIOP as a layer of larger particles (from the color ratios) with high depolarization ratios below the sulfate layer. We did not observe this in our study, but it could be important if studying the two eruptions mentioned by the reviewer. For Kelut, the sedimentation rate was small (~500 m/month) (Vernier et al., 2016) compared with the time-frame (and vertical resolution) in our study. However, studying ash rich eruptions may require further development of our method.

The relevance of the methodology for other reported eruptions could be more discussed.

We agree that this was little discussed in the manuscript. We added a few sentences on the topic in the Discussions section.

Specific comments:

Introduction: I found the comment by Xue Wu of high interest and I strongly recommend

to add their references about their similar method in the introduction.

We agree, but were not aware of their study before. We have added a few sentences on their work in the introduction section. Furthermore, we decided to contact the authors regarding their data, and have added it to the figure with comparison of vertical SO2 profiles (old Fig.7, now Fig. 8). See also comment #3 by reviewer #1.

P1 line 22: the temperature impact of the Pinatubo aerosols is debated in the community and I suggest to add more recent references such as Canty et al., Atmos. Chem. Phys.,13, 3997–4031, 2013 http://www.atmos-chem-phys.net/13/3997/2013/

We agree that our statement was somewhat exaggerated and changed to *"tens of a degree Celsius"* (citing Kremser 2016 and Canty et al., 2013)

P3 line 22: The authors should cite also the work of Günther et al. (Atmos. Chem. Phys., 18, 1217–1239, 2018https://doi.org/10.5194/acp-18-1217-2018) who present a synthesis of the emitted SO2 masses and height ranges (see their Table 1).

We have now included the SO2 profiles from Günther et al., and Höpfner et al., (2015) (referenced in Günther et al. (2018) to Fig. 8.

P4 about AIRS: Do the authors know how this dataset compares with other SO2 datasets such as IASI (Clarisse, L. et al.: Retrieval of sulphur dioxide from the infrared atmospheric sounding interferometer (IASI), Atmos. Meas. Tech., 5, 581–594, https://doi.org/10.5194/amt-5-581-2012, 2012)? If available, I would suggest to add this information in section 2.1.

We decided to limit the discussion of different SO2 instruments since our method does not use their vertical information.

Figure 1: following the general comment given above, adding the CALIOP paths superimposed to AIRS swaths would be helpful for the reader to see which part of the SO2 plume is closely captured by CALIOP. Please specify the exact date and/or time range of each swath on Figure 1.

CALIPSO/CALIOP and AQUA/AIRS ran in the same satellite constellation (A-Train). The concurrent CALIOP swaths are almost in the center of the AIRS swaths. We used 75 CALIOP swaths together with the AIRS dataset, and most of these did not occur at the same time as the AIRS observations used in Figure 2. Adding the CALIOP paths to the figure would not provide any information on which parts of the volcanic clouds that have been sampled by CALIOP since the $SO_2$ cloud move and change shape during this time lag. We therefore wish to keep the figure layout as simple as possible. In response to the general comment above, we added a concept figure (new Fig.1) illustrating the relation between the CALIOP and AIRS data.

Furthermore, we have added the time of each AIRS swath to the figure.

P6 line 11: why the investigation of the proportion of the SO2 plume located in the troposphere, which is by the way an interesting information, cannot be done automatically? Is it due to errors in following air masses using the trajectories in the troposphere (especially along isentropes)?

It is difficult to do this automatically with CALIOP, due to frequent presence of ice-clouds in the troposphere. The volcanic aerosol is difficult to separate from these ice-clouds and we therefore excluded the troposphere to prevent signals from ice-clouds.

P6 line 23: I am fine with the method using depolarization and colour ratios from CALIOP observations to point out the presence of ash and ice but there is a lack of information here. At least please refer to Vernier et al. (2016) who describes the method using depolarization ratio and add the information accordingly.

We agree with the reviewer. We have fused that section with the next one where Vernier et al. (2016) and further references on depolarization ratios of ice and ash are discussed (at the end of section 2.2).

P7 line 24: Judging by the difference in vertical resolution between ERA5 (several hundreds of meters) and CALIOP (60 m) in the stratosphere and also horizontally, please specify that the meteorological fields are spatially interpolated to release trajectories from each CALIOP profile.

We released particles at specific heights in FLEXPART corresponding to the CALIOP heights. The interpolation of the ERA5 data is done within FLEXPART and not by us. This type of interpolation is standard in dispersion models.

P7 line 27: I do not really understand what the authors mean here with the 2 standard deviations (in meter and mbar) provided here. Do they correspond to the final vertical resolution of the SO2 profile?

Yes, this is the vertical resolution of the profiles with height coordinates transformed from potential temperature to geometric altitude and pressure. We have rewritten this sentence to make our point more clear.

P8 line 17: Please better define what you mean by a pixel for CALIOP. How do you obtain pixels from a smooth signal (i.e. from fig 4b to 4d)? This is an important step I think.

We have changed the word pixel to grid-cell here.

P9 line 2: "backscatter" instead of "scattering" OK, we changed accordingly.

P9 line 4: When mentioning "95,000" particles, do you mean in total for one subcloud or for each pixel? What is the time step of the particle release? Every 6 hours?

We mean maximum 95 000 particles in total per CALIOP swath (per simulation), i.e. all pixels and sub-clouds identified in that swath. However, two of the CALIOP swaths have been divided into two Flexpart simulations when the sub-clouds were far away from each other. We release particles only once per Flexpart simulation, at the time of the CALIOP swath (or the closest half hour mark).

We have made changes to this section to clarify this.

P10 line 30: the authors do not provide details about the role of the weighting of FLEXPART outputs and how they do it. Please clarify.

We have written a more clear explanation on how the weighting was performed.

P11 line 3: I do not really agree with this statement. E-folding of SO2 is about 13-17 days

(see Haywood et al., 2010; Lurton et al., 2018). It can be mentioned here that 9 days is

lower than the reported e-foldings for the Sarychev proving that SO2 is still present in high

quantities over the 9-day time length of the trajectories.

We agree that the transformation can have some impact on our analysis. It is difficult to estimate this impact since the e-folding time for SO2 is uncertain. We have changed the text in section 3.3 to:
*"…The time it takes for the aerosol to form could thus affect the representation of the cloud. The nine days used here is shorter than previous estimates of volcanic SO$_2$ conversion in the stratosphere (Andersson et al., 2013), but ongoing SO$_2$ conversion adds uncertainty to our estimation. The short time span covered in the present study can thus be assumed to have a small effect on β$_{aer}$ per SO$_2$. This can be seen in Fig. 8 in Friberg et al. (2018), where the stratospheric aerosol load from Sarychev peaks in September, i.e. months after the last swath used in this study…"*

Figure 5: I suggest the background of the AIRS swaths to be coloured in grey rather than

dark blue to better highlight SO2 fields.

Setting a threshold for the background SO2 concentration is difficult. We have used the same color scale for the swaths in Figures 1, 5, and 6, and wish to keep it that way.

Did the authors focus on the 18th? Is it because there is concentrated SO2?

Yes, we focused on the 18$^{th}$ since AIRS had good coverage of all volcanic sub-clouds during that day. That way we could obtain an integrated view of all the Sarychev eruptions taking place in June 2009.

Why trajectories are calculated over 9 days? .

Our wish is to use as many CALIOP swaths as possible to get the best possible representation of the vertical distributions in all sub-clouds.

In figure 5, how many CALIOP profiles have generated each mapping of transported aerosols and matched with AIRS swaths? If I understand well, all trajectories over the 14-22 June period have been used to reconstruct one single AIRS swath but why not focussing on a day-by-day basis, i.e. considering only trajectory calculations from the CALIOP tracks on the same day of the AIRS swaths? The number of CALIOP profiles used to generate the vertical profile of each AIRS swath is given in the figure caption for figure 6. The different CALIOP swaths capture different parts of the sub-clouds. Hence, using CALIOP swaths of several days, instead of only one day, leads to better coverage of the SO2 clouds in the AIRS swaths. This is illustrated in the new concept figure (Fig. 1). This would possibly limit effects of aerosol sedimentation (as a result of growth and coalescence) that can bias the mapping if trajectories are calculated on a too long period. Sedimentation rates of fine mode particles are too short to influence our data in the time-frame and vertical resolution used here (Please see comments above).

Figure 6: Not all the labels are visible on fig. 6a because of the colour choice. 11 AIRS swaths are labelled in fig. 6a but only 9 are shown on the top left list. For figs. 6b, c, d I am wondering if using a log-scale will make all profiles corresponding to the AIRS swaths in fig. 6a more visible.

We did not compile vertical profiles for those swaths, since the volcanic SO2 concentration was too low and no CALIOP data could be matched with these AIRS swaths. We have now added this information to the figure caption. We changed the label fonts to make it more accessible to the reader.

The authors have chosen the 18-19 June for the application of their method. However, at this stage, the SO2 plume is already geographically extended. Then, does the method properly capture the more localized and main injection event (i.e. June 15th in Haywood et al. and Lurton et al. studies)? This is of primary importance for robust initialization of models that account for SO2 chemical cycle producing sulphate particles.

Fig.2 shows all the eruptions from Sarychev, where different eruptions usually are found at different geographical locations. No $SO_2$ clouds are found outside the $SO_2$ swaths from AIRS shown in the timeframe of Fig. 2. This figure is the central time of the computations, containing all the Sarychev eruptions that have taken place before the 18-19 June, including June 15th. By using data from several days that we connect to different 18-19 June AIRS swaths by trajectory computations, we obtain a large number of profiles describing the different eruptions. That way the number of horizontally narrow, but vertically highly resolved CALIOP observations are increased. By this approach e.g. the AIRS swath containing most $SO_2$ was intersected by 34 CALIOP profiles (caption of Fig. 6).

All sub-clouds could not be fully captured before the 18[th]. Furthermore, observations of fresh dense SO2 clouds may result in saturation of the signal and underestimation of the SO2 mass so it is not necessarily a better approach to use data directly after the event.

Figure 7: I suggest also to indicate the date of the profiles in the mentioned studies.

We have added the dates of the other profiles to the figure caption.

P14 lines 12-13: I am not sure that if the model studies indeed missed the highest cloud of SO2 (i.e. around 15.5 km in fig.7) the consequence on the retrieved aerosol space-time distribution is significant. In Lurton et al. for instance, the agreement between the simulated aerosol profiles and in

situ observations are matching pretty well even if the initial SO2 injection is underestimated by considering the results in fig. 7 (red profile). This may be due to the coarse vertical resolution of global models in the lower stratosphere (~1km) which dilutes (or spreads out) the vertical distribution of the SO2 profile and limits the impact on the vertical profile of the subsequently produced aerosols. Also, the model results tend to show longer residence times than in the observations. I suggest the authors to mention this possibility.

We agree that it is important to mention this and have added the following sentence to the manuscript."...Nevertheless, the effects of missing the highest $SO_2$ clouds depend on the vertical resolution of the models in the stratosphere and how well they capture the circulation there. "

I would suggest to remove the sentence ("Their release...") We agree with the reviewer, and changed.

P15 lines 27-29: "Our deduced...sulphate particles." I suggest to remove or modify this statement since 1) the SO2 vertical distribution (which is visible only over the first weeks after the eruption) cannot be directly connected to the one of sulphate particles several months after the eruption as a result of vertical motion (Brewer-Dobson Circulation, sedimentation) sustained by the particles over such a long period

We believe that our statement is correct and wish to keep it with some modifications. Sarychev exploded in the season with minimum subsidence from the stratosphere and aerosol was mixed to the tropics in a shallow BD branch, prolonging the influence of the eruption. The Sarychev aerosol did not reach the upper BD-branch and were thus not subject to upwelling in that branch. Furthermore, it is evident from the figure 8 in Friberg 2018 that more than half of the volcanic aerosol reached above the 380 K isentrope. This is seen already in July, and the shallow BD-branch holds more than half of the volcanic aerosol load until December (when the stratospheric subsidence maximizes (Appenzeller et al. 1996). We decided to add information to the manuscript on this in both the discussion and conclusions sections, clarifying that half of the volcanic aerosol was located above the 380 K isentrope already in July. and 2) no modelling study has been conducted yet to quantify the effect of the new vertical profile of SO2 after the publication of the authors'work. We agree that modelling would be valuable. In an upcoming study, we will simulate aerosol formation and climate response with ours, and others, vertical SO2 profiles.

References:

Appenzeller C. , Holton J. R. , Rosenlof K. H . Seasonal variation of mass transport across the tropopause. J. Geophys. Res. 1996; 101: 15071–15078.

 Fairlie, T. D., Vernier, J.-P., Natarajan, M., and Bedka, K. M.: Dispersion of the Nabro volcanic plume and its relation to the Asian summer monsoon, Atmos. Chem. Phys., 14, 7045–7057, https://doi.org/10.5194/acp-14-7045-2014, 2014.

---

## Author Response (AR2)

1) The addition of your new figure 1 made the method more clear, yet the figure was not sufficiently described in the text. If I understand correctly, the thick, solid segments on the CALIOP orbital tracks (do not call them "swaths"!) are the segments where volcanic aerosol was detected, right? These are then selected for further analysis, i.e., as starting points for trajectory analysis. The light blue cloud is presumably the SO2 plume detected during successive AIRS swaths, and the grey bits the areas where FLEXPART trajectories lead to plume altitude information, correct? Please explain this in the text. Then link Figures 1 and 4 (Sect. 3) to make the method explanation more complete We have changed the figure text according to your suggestion. We have linked it to Section 3 and to Fig. 4, and refer to the concept sketch throughout the text.

2) "tens of degrees Celsius" -> "several tenths of a degree Celsius"
We changed accordingly.

3) Your method yields results that differ markedly from all other data in Fig. 8. Is there evidence that your method is better than the others? Please explain in more detail why the results are so different, as simply stating that the other results are based on different data sets is not sufficient. Why did Wu et al. not see the two peaks and why do they strongly underestimate the total amount of SO2? The Haywood and Mills profiles seem to be much closer to what you found than Höpfner or Ge. How does that influence model results? This is actually a very important point, because if you want modelers to use your computationally expensive method for investigations of the (radiative) effects of volcanic eruptions, you'll need to convince them that it is worth it. So: what do you think is the improvement that can be achieved by using your method or data?

We have rewritten two paragraphs in the Discussions section. We now discuss all datasets and included a more thorough discussion on our dataset. We also clarified that simulations with higher vertical resolution should result in more realistic model simulations.

Wu et al. had only slightly lower SO2 mass in the stratosphere than we have (0.9 Tg vs. 1.1 Tg). They don't see any big peaks since their methodology differs markedly from ours. They use a trajectory model to infer the altitude distribution by computing trajectories at all altitudes from 0 - 20 km to find out which trajectories coincide with the measured horizontal extension of an SO2 cloud. That could lead to false positives, i.e. that altitude ranges not affected by volcanism could be transported in the same way as the volcanic layer, thus causing a broadening. In contrast we used altitude distribution from a lidar with 60 m vertical resolution the altitude range studied, which is about the best than can be obtained. Another difference is that Wu et al used altitude as their vertical parameter, whereas we used potential temperature. The relation between altitude and potential temperature varies over time and space, and the fact that transport in the stratosphere (in the first approximation) occur along potential temperature surfaces could cause additional broadening of the Wu et al profile. Wu et al., finally, validated their profile with an aerosol signal from the MIPAS instrument having, for the present purpose, the poor vertical resolution of 3 - 4 km.

When these issues have been resolved to my satisfaction, I will gladly accept the manuscript for publication in AMT.